# Reputation-based Worker Filtering in Crowdsourcing

**Srikanth Jagabathula**[1] **Lakshminarayanan Subramanian**[2,3] **Ashwin Venkataraman**[2,3]

[1]Department of IOMS, NYU Stern School of Business
[2]Department of Computer Science, New York University
[3]CTED, New York University Abu Dhabi
sjagabat@stern.nyu.edu   {lakshmi,ashwin}@cs.nyu.edu

## Abstract

In this paper, we study the problem of aggregating noisy labels from crowd workers to infer the underlying true labels of binary tasks. Unlike most prior work which has examined this problem under the random worker paradigm, we consider a much broader class of *adversarial* workers with no specific assumptions on their labeling strategy. Our key contribution is the design of a computationally efficient reputation algorithm to identify and filter out these adversarial workers in crowdsourcing systems. Our algorithm uses the concept of optimal semi-matchings in conjunction with worker penalties based on label disagreements, to assign a reputation score for every worker. We provide strong theoretical guarantees for deterministic adversarial strategies as well as the extreme case of *sophisticated* adversaries where we analyze the worst-case behavior of our algorithm. Finally, we show that our reputation algorithm can significantly improve the accuracy of existing label aggregation algorithms in real-world crowdsourcing datasets.

## 1   Introduction

The growing popularity of online crowdsourcing services (e.g. Amazon Mechanical Turk, Crowd-Flower etc.) has made it easy to collect low-cost labels from the crowd to generate training datasets for machine learning applications. However, these applications remain vulnerable to noisy labels introduced either unintentionally by unreliable workers or intentionally by spammers and malicious workers [10, 11]. Recovering the underlying true labels in the face of noisy input in online crowd-sourcing environments is challenging due to three key reasons: (a) Workers are often anonymous and transient and can provide random or even malicious labels (b) The reliabilities or reputations of the workers are often unknown (c) Each task may receive labels from only a (small) subset of the workers.

Several existing approaches aim to address the above challenges under the following standard setup. There is a set $\mathcal{T}$ of binary tasks, each with a true label in $\{-1, 1\}$. A set of workers $W$ are asked to label the tasks, and the assignment of the tasks to the workers can be represented by a bipartite graph with the workers on one side, tasks on the other side, and an edge connecting each worker to the set of tasks she is assigned. We term this the *worker-task assignment* graph. Workers are assumed to generate labels according to a probabilistic model - given a task $t$, a worker $w$ provides the true label with probability $p_w$. Note that every worker is assumed to label each task *independent* of other tasks. The goal then is to infer the underlying true labels of the tasks by aggregating the labels provided by the workers. Prior works based on the above model can be broadly classified into two categories: machine-learning based and linear-algebra based. The machine-learning approaches are typically based on variants of the EM algorithm [3, 16, 24, 14]. These algorithms perform well in most scenarios, but they lack any theoretical guarantees. More recently, linear-algebra based algorithms [9, 6, 2] have been proposed, which provide guarantees on the error in estimating the true labels of the tasks (under appropriate assumptions), and have also been shown to perform well on various real-world datasets. While existing work focuses on workers making random errors, recent work and anecdotal evidence have shown that worker labeling strategies that are common in practice do not fit the standard random model [19]. Specific examples include vote pollution attacks

on Digg [18], malicious behavior in social media [22, 12] and low-precision worker populations in crowdsourcing experiments [4].

In this paper, we aim to go beyond the standard random model and study the problem of inferring the true labels of tasks under a much broader class of *adversarial* worker strategies with no specific assumptions on their labeling pattern. For instance, deterministic labeling, where the workers always give the same label, cannot be captured by the standard random model. Also, malicious workers can employ arbitrary labeling patterns to degrade the accuracy of the inferred labels. Our goal is to accurately infer the true labels of the tasks without restricting workers' strategies.

**Main results.** Our main contribution is the design of a reputation algorithm to identify and filter out adversarial workers in online crowdsourcing systems. Specifically, we propose 2 computationally efficient algorithms to compute worker reputations using *only* the labels provided by the workers (see Algorithms 1 and 2), which are robust to manipulation by adversaries. We compute worker reputations by assigning penalties to a worker for each task she is assigned. The assigned penalty is higher for tasks on which there is "a lot" of disagreement with the other workers. The penalties are then aggregated in a "load-balanced" manner using the concept of *optimal semi-matchings* [7]. The reputation algorithm is designed to be used in conjunction with any of the existing label aggregation algorithms that are designed for the standard random worker model: workers with low reputations[1] are filtered out and the aggregation algorithm is used on the remaining labels. As a result, our algorithm can be used to boost the performance of existing label aggregation algorithms.

We demonstrate the effectiveness of our algorithm using a combination of strong theoretical guarantees and empirical results on real-world datasets. Our analysis considers three scenarios. First, we consider the standard setting in which workers are not adversarial and provide labels according to the random model. In this setting, we show that when the worker-task assignment graph is $(l, r)$-regular, the reputation scores are proportional to the reliabilities of the workers (see Theorem 1), so that only unreliable workers are filtered out. As a result, our reputation scores are consistent with worker reliabilities in the absence of adversarial workers. The analysis becomes significantly complicated for more general graphs (a fact observed in prior works; see [2]); hence, we demonstrate improvements using simulations and experiments on real world datasets. Second, we evaluate the performance of our algorithm in the presence of workers who use deterministic labeling strategies (always label 1 or −1). For these strategies, when the worker-task assignment graph is $(l, r)$-regular, we show (see Theorem 2) that the adversarial workers receive lower reputations than their "honest" counterparts, provided honest workers have "high enough" reliabilities – the exact bound depends on the prevalence of tasks with true label 1, the fraction of adversarial workers and the average reliability of the honest workers.

Third, we consider the case of *sophisticated* adversaries, i.e. worst-case adversarial workers whose goal is to maximize the number of tasks they affect (i.e. cause to get incorrect labels). Under this assumption, we provide bounds on the "damage" they can do: We prove that irrespective of the label aggregation algorithm (as long as it is agnostic to worker/task identities), there is a non-trivial minimum fraction of tasks whose true label is incorrectly inferred. This bound depends on the graph structure of the honest worker labeling pattern (see Theorem 3 for details). Our result is valid across different labeling patterns and a large class of label aggregation algorithms, and hence provides *fundamental* limits on the damage achievable by adversaries. Further, we propose a label aggregation algorithm utilizing the worker reputations computed in Algorithm 2 and prove the existence of an upper bound on the worst-case accuracy in inferring the true labels (see Theorem 4). Finally, using several publicly available crowdsourcing datasets (see Section 4), we show that our reputation algorithm: (a) can help in enhancing the accuracy of state-of-the-art label aggregation algorithms (b) is able to detect workers in these datasets who exhibit certain 'non-random' strategies.

**Additional Related Work:** In addition to the references cited above, there have been works which use gold standard tasks, i.e. tasks whose true label is already known [17, 5, 11] to correct for worker bias. [8] proposed a way of quantifying worker quality by transforming the observed labels into soft posterior labels based on the estimated *confusion matrix* [3]. The authors in [13] propose an empirical Bayesian algorithm to eliminate workers who label randomly without looking at the particular task (called *spammers*), and estimate the consensus labels from the remaining workers. Both these

works use the estimated parameters to define "good workers" whereas we compute the reputation scores using only the labels provided by the workers. The authors in [20] focus on detecting specific kinds of spammers and then replace their labels with new workers. We consider all types of adversarial workers, not just spammers and don't assume access to a pool of workers who can be asked to label the tasks.

## 2  Model and reputation algorithms

**Notation.** Consider a set of tasks $\mathcal{T}$ having true labels in $\{1, -1\}$. Let $y_j$ denotes the true label of a task $t_j \in \mathcal{T}$ and suppose that the tasks are sampled from a population in which the *prevalence* of the positive tasks is $\gamma \in [0, 1]$, so that there is a fraction $\gamma$ of tasks with true label 1. A set of workers $W$ provide binary labels to the tasks in $\mathcal{T}$. We let $G$ denote the bipartite *worker-task assignment* graph where an edge between worker $w_i$ and task $t_j$ indicates that $w_i$ has labeled $t_j$. Further, let $w_i(t_j)$ denote the label provided by worker $w_i$ to task $t_j$, where we set $w_i(t_j) = 0$ if worker $w_i$ did not label task $t_j$. For a task $t_j$, let $W_j \subset W$ denote the set of workers who labeled $t_j$ and likewise, for a worker $w_i$, let $\mathcal{T}_i$ denote the set of tasks the worker has labeled. Denote by $d_j^+$ (resp. $d_j^-$) the number of workers labeling task $t_j$ as 1 (resp. $-1$). Finally, let $\mathcal{L} \in \{1, 0, -1\}^{|W| \times |\mathcal{T}|}$ denote the *matrix* representing the labels assigned by the workers to the tasks, i.e. $\mathcal{L}_{ij} = w_i(t_j)$. Given $\mathcal{L}$, the goal is to infer the true labels $y_j$ of the tasks.

**Worker model.** We consider the setting in which workers may be *honest* or *adversarial*. That is, $W = H \cup A$ with $H \cap A = \emptyset$. Honest workers are assumed to provide labels according to a probabilistic model: for task $t_j$ with true label $y_j$, worker $w_i$ provides label $y_j$ with probability $p_i$ and $-y_j$ with probability $1 - p_i$. Note that the parameter $p_i$ doesn't depend on the particular task that the worker is labeling, so an honest worker labels each task independently. It is standard to define the *reliability* of an honest worker as $\mu_i = 2p_i - 1$, so that we have $\mu_i \in [-1, 1]$. Further, we assume that the honest workers are sampled from a population with average reliability $\mu > 0$. Adversaries, on the other hand, may adopt any arbitrary (deterministic or randomized) labeling strategy that cannot be described using the above probabilistic process. For instance, the adversary could always label all tasks as 1, irrespective of the true label. Another example is when the adversary decides her labels based on existing labels cast by other workers (assuming the adversary has access to such information). Note however, that adversarial workers need not always provide the incorrect labels. Essentially, the presence of such workers breaks the assumptions of the model and can adversely impact the performance of aggregation algorithms. Hence, our focus in this paper is on designing algorithms to identify and filter out such adversarial workers. Once this is achieved, we can use existing state-of-the-art label aggregation algorithms on the remaining labels to infer the true labels of the tasks.

To identify these adversarial workers, we propose an algorithm for computing "reputation" or "trust" scores for each worker. More concretely, we assign penalties (in a suitable way) to every worker and higher the penalty, worse the reputation of the worker. First note that any task that has all 1 labels (or $-1$ labels) does not provide us with any information about the reliabilities of the workers who labeled the task. Hence, we focus on the tasks that have both 1 and $-1$ labels and we call this set the *conflict set* $\mathcal{T}_{cs}$. Further, since we have no "side" information on the identities of workers, any reputation score computation must be based solely on the labels provided by the workers.

We start with the following basic idea to compute reputation scores: a worker is penalized for every 'conflict' s/he is involved in (a task in the conflict set the worker has labeled on). This idea is motivated by the fact that in an ideal scenario, where all honest workers have a reliability $\mu_i = 1$, a conflict indicates the presence of an adversary and the reputation score aims to capture a measure of the number of conflicts each worker is involved in: the higher the number of conflicts, the worse the reputation score. However, a straightforward aggregation of penalties for each worker may over-penalize (honest) workers who label several tasks.

In order to overcome the issue of over-penalizing (honest) workers, we propose two techniques: (a) *soft* and (b) *hard* assignment of penalties. In the *soft* assignment of penalties (Algorithm 1), we assign a penalty of $1/d_j^+$ to all workers who label 1 on task $t_j$ and $1/d_j^-$ to all workers who label $-1$ on task $t_j$. Then, for each worker, we aggregate the penalties across all assigned tasks by taking the average. The above assignment of penalties implicitly rewards agreements by making the penalty inversely proportional to the number of other workers that agree with a worker. Further, taking the average normalizes for the number of tasks labeled by the worker. Since we expect the

honest workers to agree with the majority more often than not, we expect this technique to assign lower penalties to honest workers when compared to adversaries. The soft assignment of penalties can be shown to perform quite well in identifying low reliability and adversarial workers (refer to Theorems 1 and 2). However, it may still be subject to manipulation by more "sophisticated" adversaries who can adapt and modify their labeling strategy to target certain tasks and to inflate the penalty of specific honest workers. In fact for such worst-case adversaries, we can show that (Theorem 3) given any honest worker labeling pattern, there exists a lower bound on the number of tasks whose true label cannot be inferred correctly, by *any* label aggregation algorithm.

To address the case of these sophisticated adversaries, we propose a *hard* penalty assignment scheme (Algorithm 2) where the key idea is not to distribute the penalty evenly across all workers; but to only choose two workers to penalize per conflict task: one "representative" worker among those who labeled 1 and another "representative" worker among those who labeled $-1$. While choosing such workers, the goal is to pick these representative workers in a load-balanced manner to "spread" the penalty across all workers, so that it is not concentrated on one/few workers. The final penalty of each worker is the sum of the accrued penalties across all the (conflict) tasks assigned to the worker. Intuitively, such hard assignment of penalties will penalize workers with higher degrees and many conflicts (who are potential 'worst-case' adversaries), limiting their impact.

We use the concept of *optimal semi-matchings* [7] on bipartite graphs to distribute penalties in a load balanced manner, which we briefly discuss here. For a bipartite graph $B = (U, V, E)$, a *semi-matching* in $B$ is a set of edges $M \subseteq E$ such that each vertex in $V$ is incident to exactly one edge in $M$ (note that vertices in $U$ could be incident to multiple edges in $M$). A semi-matching generalizes the notion of matchings on bipartite graphs. To define an optimal semi-matching, we introduce a cost function for a semi-matching - for each $u \in U$, let $deg_M(u)$ denote the number of edges in $M$ that are incident to $u$ and let $cost_M(u)$ be defined as $cost_M(u) = \sum_{i=1}^{deg_M(u)} i = \frac{deg_M(u)(deg_M(u)+1)}{2}$. An *optimal semi-matching* then, is one which minimizes $\sum_{u \in U} cost_M(u)$. This notion of cost is motivated by the load balancing problem for scheduling tasks on machines (refer to [7] for further details). Intuitively, an optimal semi-matching *fairly* matches the $V$-vertices across the $U$-vertices so that the maximum "load" on any $U$-vertex is minimized.

| **Algorithm 1** SOFT PENALTY | **Algorithm 2** HARD PENALTY |
|---|---|
| 1: **Input:** $W$, $\mathcal{T}$ and $\mathcal{L}$ <br> 2: For every task $t_j \in \mathcal{T}_{cs}$, assign penalty $s_{ij}$ to each worker $w_i \in W_j$ as follows: <br> $\quad s_{ij} = \frac{1}{d_j^+}$ if $\mathcal{L}_{ij} = 1$ <br> $\quad s_{ij} = \frac{1}{d_j^-}$ if $\mathcal{L}_{ij} = -1$ <br> 3: **Output:** Penalty of worker $w_i$ <br><br> $\quad pen(w_i) = \frac{\sum_{t_j \in \mathcal{T}_i \cap \mathcal{T}_{cs}} s_{ij}}{\|\mathcal{T}_i \cap \mathcal{T}_{cs}\|}$ | 1: **Input:** $W$, $\mathcal{T}$ and $\mathcal{L}$ <br> 2: Create a bipartite graph $\mathcal{B}^{cs}$ as follows: <br> (i) Each worker $w_i \in W$ is represented by a node on the left (ii) Each task $t_j \in \mathcal{T}_{cs}$ is represented by two nodes on the right $t_j^+$ and $t_j^-$ (iii) Add the edge $(w_i, t_j^+)$ if $\mathcal{L}_{ij} = 1$ or edge $(w_i, t_j^-)$ if $\mathcal{L}_{ij} = -1$. <br> 3: Compute an optimal semi-matching OSM on $\mathcal{B}^{cs}$ and let $d_i \Leftarrow$ degree of worker $w_i$ in OSM <br> 4: **Output:** Penalty of worker $w_i$ $pen(w_i) = d_i$ |

## 3 Theoretical Results

**Soft penalty.** We focus on $(l, r)$-regular worker-task assignment graphs in which every worker is assigned $l$ tasks and every object is labeled by $r$ workers. The performance of our reputation algorithms depend on the reliabilities of the workers as well as the true labels of the tasks. Hence, we consider the following probabilistic model: for a given $(l, r)$-regular worker-task assignment graph $G$, the reliabilities of the workers and the true labels of tasks are sampled independently (from distributions described in Section 2). We then analyze the performance of our algorithms as the task degree $r$ (and hence number of workers $|W|$) goes to infinity. Specifically, we establish the following results (the proofs of all theorems are in the supplementary material).

**Theorem 1.** *Suppose there are no adversarial workers, i.e $A = \emptyset$ and that the worker-task assignment graph $G$ is $(l, r)$-regular. Then, with high probability as $r \to \infty$, for any pair of workers $w_i$ and $w_{i'}$, $\mu_i < \mu_{i'} \implies pen(w_i) > pen(w_{i'})$, i.e. higher reliability workers are assigned lower penalties by Algorithm 1.*

The probability in the above theorem is according to the model described above. Note that the theorem justifies our definition of the reputation scores by establishing their consistency with worker reliabilities in the absence of adversarial workers. Next, consider the setting in which adversarial workers adopt the following *uniform strategy*: label 1 on all assigned tasks (the $-1$ case is symmetric).

**Theorem 2.** *Suppose that the worker-task assignment graph $G$ is $(l, r)$-regular. Let the probability of an arbitrary worker being honest be $q$ and suppose each adversary adopts the uniform strategy in which she labels $1$ on all the tasks assigned to her. Denote an arbitrary honest worker as $h_i$ and any adversary as $a$. Then, with high probability as $r \to \infty$, we have*

1. *If $\gamma = \frac{1}{2}$ and $\mu_i = 1$, then $pen(h_i) < pen(a)$ if and only if $q > \frac{1}{1+\mu}$*

2. *If $\gamma = \frac{1}{2}$ and $q > \frac{1}{1+\mu}$, then $pen(h_i) < pen(a)$ if and only if*

$$\mu_i \geq \frac{(2-q)(1-q-q^2\mu^2) - q^2\mu^2}{(2-q)q + q^2\mu^2}$$

The above theorem establishes that when adversaries adopt the uniform strategy, the soft-penalty algorithm assigns lower penalties to honest workers whose reliability excess a threshold, as long as the fraction of honest workers is "large enough". Although not stated, the result above immediately extends (with a modified lower bound for $\mu_i$) to the case when $\gamma > 1/2$, which corresponds to adversaries adopting *smart* strategies by labeling based on the prevalence of positive tasks.

**Sophisticated adversaries**. Numerous real-world incidents show evidence of malicious worker behavior in online systems [18, 22, 12]. Moreover, attacks on the training process of ML models have also been studied [15, 1]. Recent work [21] has also shown the impact of powerful adversarial attacks by administrators of crowdturfing (malicious crowdsourcing) sites. Motivated by these examples, we consider sophisticated adversaries:

**Definition 1.** *Sophisticated adversaries are computationally unbounded and colluding. Further, they have knowledge of the labels provided by the honest workers and their goal is to maximize the number of tasks whose true label is incorrectly identified.*

We now raise the following question: In the presence of sophisticated adversaries, does there exist a *fundamental limit* on the number of tasks whose true label can be correctly identified, irrespective of the aggregation algorithm employed to aggregate the worker labels?

In order to answer the above question precisely, we introduce some notation. Let $n = |W|$ and $m = |\mathcal{T}|$. Then, we represent any label aggregation algorithm as a *decision rule* $R : \mathcal{L} \to \{1, -1\}^m$, which maps the observed labeling matrix $\mathcal{L}$ to a set of output labels for each task. Because of the absence of any auxiliary information about the workers or the tasks, the class of decision rules, say $\mathcal{C}$, is invariant to permutations of the identities of workers and/or tasks. More precisely, $\mathcal{C}$ denotes the class of decision rules that satisfy $R(P\mathcal{L}Q) = R(\mathcal{L})Q$, for any $n \times n$ permutation matrix $P$ and $m \times m$ permutation matrix $Q$. We say that a task is *affected* if the decision rule outputs the incorrect label for the task. We define the *quality* of a decision rule $R(\cdot)$ as the *worst-case* number of affected tasks over all possible true labelings and adversary strategies with a *fixed* honest worker labeling pattern. Fixing the honest worker labeling pattern allows isolation of the effect of the adversary strategy on the accuracy of the decision rule. Considering the worst-case over all possible true labels makes the metric robust to ground-truth assignments, which are typically application specific.

Next to formally define the quality, let $\mathcal{B}_H$ denote the *honest worker-task assignment* graph and $\mathbf{y} = (y_1, y_2, \ldots, y_m)$ denote the vector of true labels for the tasks. Note that since the number of affected tasks also depends on the actual honest worker labels, we further assume that all honest workers have reliability $\mu_i = 1$, i.e they always label correctly. This assumption allows us to attribute any mis-identification of true labels to the presence of adversaries because, otherwise, in the absence of any adversaries, the true labels of all the tasks can be trivially identified. Finally, let $\mathcal{S}_k$ denote the strategy space of $k$ adversaries, where each strategy $\sigma \in \mathcal{S}_k$ specifies the $k \times m$ label matrix provided by the adversaries. Since we do not restrict the adversary strategy in any way, it follows that $\mathcal{S}_k = \{-1, 0, 1\}^{k \times m}$. The *quality* of a decision rule is then defined as

$$\text{Aff}(R, \mathcal{B}_H, k) = \max_{\sigma \in \mathcal{S}_k, \mathbf{y} \in \{1, -1\}^m} \left| \left\{ t_j \in \mathcal{T} : R_{t_j}^{\mathbf{y}, \sigma} \neq y_j) \right\} \right|,$$

where $R_t^{\mathbf{y},\sigma} \in \{1,-1\}$ is the label output by the decision rule $R$ for task $t$ when the true label vector is $\mathbf{y}$ and the adversary strategy is $\sigma$. Note that our notation $\mathrm{Aff}(R, \mathcal{B}_H, k)$ makes the dependence of the quality measure on the honest worker-task assignment graph $\mathcal{B}_H$ and the number of adversaries $k$ explicit. We answer the question raised above in the *affirmative*, i.e. there does exist a fundamental limit on identification. In the theorem below, $\mathrm{PreIm}(\mathcal{T}')$ is the set of *honest* workers who label atleast one task in $\mathcal{T}'$.

**Theorem 3.** *Suppose that $k = |A|$ and $\mu_h = 1$ for all honest workers $h \in H$. Then, given any honest worker-task assignment graph $\mathcal{B}_H$, there exists an adversary strategy $\sigma^* \in \mathcal{S}_k$ that is independent of any decision rule $R \in \mathcal{C}$ such that*

$$L \leq \max_{\mathbf{y} \in \{-1,1\}^m} \mathrm{Aff}(R, \sigma^*, \mathbf{y}) \ \ \forall R \in \mathcal{C}, \quad where$$

$$L = \frac{1}{2} \max_{\mathcal{T}' \subseteq \mathcal{T} \,:\, |\mathrm{PreIm}(\mathcal{T}')| \leq k} |\mathcal{T}'|,$$

*and $\mathrm{Aff}(R, \sigma^*, \mathbf{y})$ denotes the number of affected tasks under adversary strategy $\sigma^*$, decision rule $R$, and true label vector $\mathbf{y}$ (with the assumption that max over an empty set is zero).*

We describe the main idea of the proof which proceeds in two steps: (i) we provide an explicit construction of an adversary strategy $\sigma^*$ that depends on $\mathcal{B}_H$ and $\mathbf{y}$, and (ii) we show the existence of another true labeling $\hat{\mathbf{y}}$ such that $R$ outputs exactly the same labels in both scenarios. The adversary labeling strategy we construct uses the idea of *indistinguishability*, which captures the fact that by carefully choosing their labels, the adversaries can render themselves indistinguishable from honest workers. Specifically, in the simple case when there is only one honest worker, the adversary simply labels the opposite of the honest worker on all assigned tasks, so that each task has two labels of opposite parity. It can be argued that since there is no other information, it is impossible for any decision rule $R \in \mathcal{C}$ to distinguish the honest worker from the adversary and hence identify the true label of any task (better than a random guess). We extend this to the general case, where the adversary "targets" atmost $k$ honest workers and derives a strategy based on the subgraph of $\mathcal{B}_H$ restricted to the targeted workers. The resultant strategy can be shown to result in incorrect labels for atleast $L$ tasks for some true labeling of the tasks.

**Hard penalty**. We now analyze the performance of the hard penalty-based reputation algorithm in the presence of *sophisticated adversarial workers*. For the purposes of the theorem, we consider a natural extension of our reputation algorithm to also perform label aggregation (see figure 1).

**Theorem 4.** *Suppose that $k = |A|$ and $\mu_i = 1$ for each honest worker, i.e an honest worker always provides the correct label. Further, let $d_1 \geq d_2 \geq \cdots \geq d_{|H|}$ denote the degrees of the honest workers in the optimal semi-matching on $\mathcal{B}_H$. For any true labeling of the tasks and under the penalty-based label aggregation algorithm (with the convention that $d_i = 0$ for $i > |H|$):*

1. *There exists an adversary strategy $\sigma^*$ such that the number of tasks affected $\geq \sum_{i=1}^{k-1} d_i$.*
2. *No adversary strategy can affect more than $U$ tasks where*
    (a) *$U = \sum_{i=1}^{k} d_i$, when atmost one adversary provides correct labels*
    (b) *$U = \sum_{i=1}^{2k} d_i$, in the general case*

A few remarks are in order. First, it can be shown [7] that for optimal semi-matchings, the degree sequence is unique and therefore, the bounds in the theorem above are uniquely defined given $\mathcal{B}_H$. Also, the assumption that $\mu_i = 1$ is required for analytical tractability, proving theoretical guarantees in crowd-sourced settings (even without adversaries) for general graph structures is notoriously hard [2]. Note that the result of Theorem 4 provides both a lower and upper bound for the number of tasks that can be affected by $k$ adversaries when using the penalty-based aggregation algorithm. The characterization we obtain is reasonably tight for the case when atmost 1 adversary provides correct labels; in this case the gap between the upper and lower bounds is $d_k$, which can be "small" for $k$ large enough. However, our characterization is loose in the general case when adversaries can label arbitrarily; here the gap is $\sum_{i=k}^{2k} d_i$ which we attribute to our proof technique and conjecture that the upper bound of $\sum_{i=1}^{k} d_i$ also applies in the more general case.

## 4 Experiments

In this section, we evaluate the performance of our reputation algorithms on both synthetic and real datasets. We consider the following popular label aggregation algorithms: (a) simple majority vot-

| | Random | | Malicious | | Uniform | | | PENALTY-BASED AGGREGATION |
|---|---|---|---|---|---|---|---|---|
| | Low | High | Low | High | Low | High | | $w_t \Leftarrow$ worker that task $t$ is mapped to in OSM in Algorithm 2 |
| MV | 9.9 | 7.9 | 16.8 | 15.6 | 26.0 | 15.0 | | **Output** $y(t) = 1$ if $d_{w_{t+}} < d_{w_{t-}}$ |
| EM | -1.9 | 6.3 | -1.6 | -49.4 | -1.2 | -9.1 | | $y(t) = -1$ if $d_{w_{t+}} > d_{w_{t-}}$ and |
| KOS | -4.3 | 13.1 | -8.3 | -98.7 | -6.5 | 12.9 | | $y(t) = 0$ otherwise |
| KOS+ | -3.9 | 7.3 | -8.3 | -69.6 | -6.0 | 10.7 | | (here $y$ refers to the label of the task |
| PRECISION | 81.7 | 82.1 | 92.5 | 59.4 | 80.8 | 62.4 | | and $d_w$ refers to the degree of worker |
| BEST | MV-SOFT | MV-HARD | MV-SOFT | KOS | MV-SOFT | MV-HARD | | $w$ in OSM) |

Figure 1: **Left**: *Percentage decrease in incorrect tasks* on synthetic data (negative indicates increase in incorrect tasks). We implemented both SOFT and HARD and report the best outcome. Also reported is the precision when removing 15 workers with the highest penalties. The columns specify the three types of adversaries and High/Low indicates the degree bias of the adversaries. The probability that a worker is honest $q$ was set to 0.7 and the prevalence $\gamma$ of positive tasks was set to 0.5. The numbers reported are an average over 100 experimental runs. The last row lists the combination with the best accuracy in each case. **Right**: The penalty-based label aggregation algorithm.

ing MV (b) the EM algorithm [3] (c) the BP-based KOS algorithm [9] and (d) KOS+, a normalized version of KOS that is amenable for arbitrary graphs (KOS is designed for random regular graphs), and compare their accuracy in inferring the true labels of the tasks, when implemented in conjunction with our reputation algorithms. We implemented iterative versions of Algorithms 1(SOFT) and 2(HARD), where in each iteration we filtered out the worker with the highest penalty and recomputed penalties for the remaining workers.

**Synthetic Dataset.** We analyzed the performance of our soft penalty-based reputation algorithm on $(l, r)$-regular graphs in section 3. In many practical scenarios, however, the worker-task assignment graph forms organically where the workers decide which tasks to label on. To consider this case, we simulated a setup of 100 workers with a power-law distribution for worker degrees to generate the bipartite worker-task assignment graph. We assume that an honest worker always labels correctly (the results are qualitatively similar when honest workers make errors with small probability) and consider three notions of adversaries: (a) *random* - who label each task 1 or $-1$ with prob. $1/2$ (b) *malicious* - who always label incorrectly and (c) *uniform* - who label 1 on all tasks. Also, we consider both cases - one where the adversaries are biased to have high degrees and the other where they have low degrees. Further, we arbitrarily decided to remove 15% of the workers with the highest penalties and we define *precision* as the percentage of workers filtered who were adversarial. Figure 1 shows the performance improvement of the different benchmarks in the presence of our reputation algorithm.

We make a few observations. First, we are successful in identifying *random* adversaries as well as low-degree *malicious* and *uniform* adversaries (precision > 80%). This shows that our reputation algorithms also perform well when worker-task assignment graphs are non-regular, complementing the theoretical results (Theorems 1 and 2) for regular graphs. Second, our filtering algorithm can result in significant reduction (upto 26%) in the fraction of incorrect tasks. In fact, in 5 out of 6 cases, the best performing algorithm incorporates our reputation algorithm. Note that since 15 workers are filtered out, labels from fewer workers are used to infer the true labels of the tasks. Despite using fewer labels, we improve performance because the high precision of our algorithms results in mostly adversaries being filtered out. Third, we note that when the adversaries are *malicious* and have high degrees, the removal of 15 workers degrades the performance of the KOS (and KOS+) and EM algorithms. We attribute this to the fact that while KOS and EM are able to automatically invert the malicious labels, we discard these labels which hurts performance, since the adversaries have high degrees. Finally, note that the SOFT (HARD) penalty algorithm tends to perform better when adversaries are biased towards low (high) degrees, and this insight can be used to aid the choice of the reputation algorithm to be employed in different scenarios.

**Real Datasets.** Next, we evaluated our algorithm on some standard datasets: (a) TREC[2]: a collection of topic-document pairs labeled as relevant or non-relevant by workers on AMT. We consider two versions: `stage2` and `task2`. (b) NLP [17]: consists of annotations by AMT workers for different NLP tasks (1) `rte` - which provides binary judgments for textual entailment, i.e. whether one

| Dataset | MV | | | EM | | | KOS | | | KOS+ | | |
|---|---|---|---|---|---|---|---|---|---|---|---|---|
| | Base | Soft | Hard | Base | Soft | Hard | Base | Soft | Hard | Base | Soft | Hard |
| rte | 91.9 | 92.1(8) | **92.5(3)** | 92.7 | 92.7 | **93.3(9)** | 49.7 | 88.8(9) | **91.6(10)** | 91.3 | 92.7(8) | **92.8(10)** |
| temp | 93.9 | 93.9 | **94.3(5)** | 94.1 | 94.1 | 94.1 | 56.9 | 69.2(4) | **93.7(3)** | 93.9 | **94.3(7)** | **94.3(1)** |
| bluebird | 75.9 | 75.9 | 75.9 | 89.8 | 89.8 | 89.8 | 72.2 | **75.9(3)** | 72.2 | 72.2 | **75.9(3)** | 72.2 |
| stage2 | 74.1 | 74.1 | **81.4(3)** | 64.7 | 65.3(6) | **78.9(2)** | 74.5 | 74.5 | **75.2(3)** | 75.5 | 76.6(2) | **77.2(3)** |
| task2 | 64.3 | 64.3 | **68.4(10)** | 66.8 | 66.8 | **67.3(9)** | 57.4 | 57.4 | **66.7(10)** | 59.3 | 59.4(4) | **67.9(9)** |
| aggregate | 80.0 | 80.0 | **82.5** | 81.6 | 81.7 | **84.7** | 62.1 | 73.2 | **79.9** | 78.4 | 79.8 | **80.9** |

Table 1: **Percentage accuracy** of benchmark algorithms when combined with our reputation algorithms. For each benchmark, the best performing combination is shown in bold. The number in the parentheses represents the number of workers filtered by our reputation algorithm (an absence indicates that no performance improvement was achieved while removing upto 10 workers with the highest penalties).

sentence can be inferred from another (2) `temp` - which provides binary judgments for temporal ordering of events. (c) `bluebird` [23] contains judgments differentiating two kinds of birds in an image. Table 1 reports the *best* accuracy achieved when upto 10 workers are filtered using our iterative reputation algorithms.

The main conclusion we draw is that our reputation algorithms are able to boost the performance of state-of-the-art aggregation algorithms by a significant amount across the datasets: the average improvement for MV and KOS+ is 2.5%, EM is 3% and for KOS is almost 18%, when using the hard penalty-based reputation algorithm. Second, we can elevate the performance of simpler algorithms such as KOS and MV to the levels of the more general (and in some cases, complicated) EM algorithm. The KOS algorithm for instance, is not only easier to implement, but also has tight theoretical guarantees when the underlying assignment graph is sparse random regular and further is robust to different initializations [9]. The modified version KOS+ can be used in graphs where worker degrees are skewed, but we are still able to enhance its accuracy. Our results provide evidence for the fact that existing random models are inadequate in capturing the behavior of workers in real-world datasets, necessitating the need for a more general approach. Third, note that the hard penalty-based algorithm outperforms the soft version across the datasets. Since the hard penalty algorithm works well when adversaries have higher degrees (a fact noticed in the simulation results above), this suggests the presence of high-degree adversarial workers in theses datasets. Finally, our algorithm was successful in identifying the following types of "adversaries": (1) *uniform* workers who always label 1 or $-1$ (in `temp`, `task2`, `stage2`), (2) *malicious* workers who mostly label incorrectly (in `bluebird`, `stage2`) and (3) *random* workers who label each task independent of its true label (such workers were identified as "spammers" in [13]). Therefore, our algorithm is able to identify a broad set of adversary strategies in addition to those detected by existing techniques.

# 5 Conclusions

This paper analyzes the problem of inferring true labels of tasks in crowdsourcing systems against a broad class of adversarial labeling strategies. The main contribution is the design of a reputation-based worker filtering algorithm that uses a combination of disagreement-based penalties and optimal semi-matchings to identify adversarial workers. We show that our reputation scores are consistent with the existing notion of worker reliabilities and further can identify adversaries that employ deterministic labeling strategies. Empirically, we show that our algorithm can be applied in real crowd-sourced datasets to enhance the accuracy of existing label aggregation algorithms. Further, we analyze the scenario of worst-case adversaries and establish lower bounds on the minimum "damage" achievable by the adversaries.

**Acknowledgments**

We thank the anonymous reviewers for their valuable feedback. Ashwin Venkataraman was supported by the Center for Technology and Economic Development (CTED).

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

## Footnotes

[1]As will become evident later, reputations are measures of how adversarial a worker is and are different from reliabilities of workers.

[2]http://sites.google.com/site/treccrowd/home
