[Supplementary Material]

# Reputation-based Worker Filtering in Crowdsourcing – Supplementary Material

## A    Proofs of the theorems

We first state a few helper lemmas.

**Lemma 1.** Suppose the graph $G$ is an $(l, r)$-regular graph, i.e. worker degree is $l$ and task degree is $r$. Then, for each $(w_i, t_j) \in G$, the following is true

$$\Pr(w_i(t_j) = 1) = \frac{1 + (2\gamma - 1)\mu}{2},$$

$$\Pr(w_i(t_j) = -1) = \frac{1 - (2\gamma - 1)\mu}{2},$$

and

$$\mathbb{E}[d_j^+] = r\frac{1 + (2\gamma - 1)\mu}{2}, \; \mathbb{E}[d_j^-] = r\frac{1 - (2\gamma - 1)\mu}{2}$$

where the probability and expectation are taken over the random process of sampling worker reliabilities, task true labels, and the labels provided by the workers.

*Proof.* The proof of this result is rather simple. We have,

$$
\begin{aligned}
\Pr(w_i(t_j) = 1 \mid \mu_i) &= \Pr(w_i(t_j) = 1 \mid y_j = 1, \mu_i)\Pr(y_j = 1 \mid \mu_i) \\
&\quad + \Pr(w_i(t_j) = 1 \mid y_j = -1, \mu_i)\Pr(y_j = -1 \mid \mu_i) \\
&= p_i\gamma + (1 - p_i)(1 - \gamma) \\
&= \gamma\frac{1 + \mu_i}{2} + (1 - \gamma)\frac{1 - \mu_i}{2} \\
&= 1/2 + (2\gamma - 1)\mu_i/2.
\end{aligned}
$$

The expression for $\Pr(w_i(t_j) = -1)$ now immediately follows from the fact that

$$\Pr(w_i(t_j) = -1) = 1 - \Pr(w_i(t_j) = 1)$$

Hence, we can write

$$
\begin{aligned}
\Pr(w_i(t_j) = 1) &= \int \Pr(w_i(t_j) = 1 \mid \mu_i)f(\mu_i)d\mu_i \\
&= \frac{1}{2} + \frac{2\gamma - 1}{2}\int \mu_i f(\mu_i)d\mu_i \\
&= 1/2 + (2\gamma - 1)\mu/2,
\end{aligned}
$$

where $\mu$ is the average reliability in the population. In a similar fashion we can write

$$
\begin{aligned}
\mathbb{E}[d_j^+] &= \sum_{i \in W_j} \Pr(w_i(t_j) = 1) \\
&= |W_j|\frac{1 + (2\gamma - 1)\mu}{2} = r\frac{1 + (2\gamma - 1)\mu}{2},
\end{aligned}
$$

where $W_j$ denotes the set of workers labeling task $t_j$ and the second equality follows from the expression for $\Pr(w_i(t_j) = 1)$ above. We have used the fact that $|W_j| = r$ because the assignment graph $G$ is $(l, r)$-regular.

The expression for $\mathbb{E}[d_j^-]$ immediately follows from the fact that $d_j^+ + d_j^- = |W_j| = r$ for any task $t_j$. The result of the lemma now follows. $\square$

**Lemma 2.** For a given realization of the worker reliabilities and task true labels, we have the following:

$$\mathbb{E}[d_j^+ \,|\, y_j = 1] = \frac{r}{2} + \frac{1}{2} \sum_{i' \in W_j} \mu_{i'}$$

$$\mathbb{E}[d_j^+ \,|\, y_j = -1] = \frac{r}{2} - \frac{1}{2} \sum_{i' \in W_j} \mu_{i'},$$

where the expectation is over the randomness in label generation. Furthermore, we must have

$$\mathbb{E}[d_j^+ \,|\, y_j = 1, w_i(t_j) = 1] = \frac{1 - \mu_i}{2} + \frac{r}{2} + \frac{1}{2} \sum_{i' \in W_j} \mu_{i'}$$

$$\mathbb{E}[d_j^+ \,|\, y_j = -1, w_i(t_j) = 1] = \frac{1 + \mu_i}{2} + \frac{r}{2} - \frac{1}{2} \sum_{i' \in W_j} \mu_{i'},$$

*Proof.* We proceed as follows.

$$\mathbb{E}[d_j^+ \,|\, y_j = 1] = \sum_{i' \in W_j} \Pr(w_{i'}(t_j) = 1 \,|\, y_j = 1) = \sum_{i' \in W_j} p_{i'} = \sum_{i' \in W_j} \frac{1 + \mu_{i'}}{2}$$

$$= \frac{|W_j|}{2} + \frac{1}{2} \sum_{i' \in W_j} \mu_{i'}$$

$$= \frac{r}{2} + \frac{1}{2} \sum_{i' \in W_j} \mu_{i'}.$$

It can be shown in a similar fashion that $\mathbb{E}[d_j^+ \,|\, y_j = -1] = r/2 - (\sum_{i' \in W_j} \mu_{i'})/2$.

Coming to the next set of equalities, we have

$$\mathbb{E}[d_j^+ \,|\, y_j = 1, w_i(t_j) = 1] = 1 + \sum_{i' \in W_j, i' \neq i} \mathbb{E}[\mathbf{1}[w_{i'}(t_j) = 1] \,|\, w_i(t_j) = 1, y_j = 1]$$

$$= 1 + \sum_{i' \in W_j, i' \neq i} \mathbb{E}[\mathbf{1}[w_{i'}(t_j) = 1] \,|\, y_j = 1]$$

$$= 1 + \sum_{i' \in W_j, i' \neq i} \Pr(w_{i'}(t_j) = 1 \,|\, y_j = 1)$$

$$= 1 + \sum_{i' \in W_j, i' \neq i} p_{i'}$$

$$= 1 - p_i + \sum_{i' \in W_j} p_{i'}$$

$$= \frac{1 - \mu_i}{2} + \sum_{i' \in W_j} p_{i'},$$

where the second inequality follows from the conditional independence of $\mathbf{1}[w_i(t_j) = 1]$ and $\mathbf{1}[w_{i'}(t_j) = 1]$ given $t_j$ for $i \neq i'$. The desired expression is then obtained by noting that $\sum_{i' \in W_j} p_{i'} = \mathbb{E}[d_j^+ \,|\, y_j = 1]$. The derivation of the expression of $\mathbb{E}[d_j^+ \,|\, y_j = -1, w_i(t_j) = 1]$ follows from a symmetric argument. The result of the lemma now follows. $\square$

## A.1 Proof of Theorem 1

For a given realization of worker reliabilities, task true labels, and worker labels to tasks, let $s_{ij}$ denote the penalty assigned by task $t_j$ to worker $w_i$. It follows from our algorithm that

$$1/s_{ij} = d_j^+ \mathbf{1}[w_i(t_j) = 1] + d_j^- \mathbf{1}[w_i(t_j) = -1]$$

$$= (d_j^+ - d_j^-)\mathbf{1}[w_i(t_j) = 1] + d_j^-$$

$$= (2d_j^+ - r)\mathbf{1}[w_i(t_j) = 1] + r - d_j^+$$

We now consider the case that $y_j = 1$. Now note that

$$\mathbb{E}[\mathbf{1}[w_i(t_j) = 1]d_j^+ \,|\, y_j = 1] = \sum_{i' \in W_j} \mathbb{E}[\mathbf{1}[w_i(t_j) = 1]\mathbf{1}[w_{i'}(t_j) = 1] \,|\, y_j = 1]$$

$$= \Pr(w_i(t_j) = 1 \,|\, y_j = 1)\left(1 + \sum_{i' \in W_j, i \neq i'} \Pr(w_{i'}(t_j) = 1 \,|\, y_j = 1)\right),$$

where we have used the fact that the random variables $\mathbf{1}[w_i(t_j) = 1]$ and $\mathbf{1}[w_{i'}(t_j) = 1]$ for $i \neq i'$ are conditionally independent given $y_j$. It then follows that (using lemmas 1 and 2 above)

$$\mathbb{E}[1/s_{ij} \,|\, y_j = 1]$$
$$= \left(2\mathbb{E}[d_j^+ \,|\, y_j = 1, w_i(t_j) = 1] - r\right)\Pr(w_i(t_j) = 1 \,|\, y_j = 1)$$
$$+ r - \mathbb{E}[d_j^+ \,|\, y_j = 1]$$
$$= \left(1 - \mu_i + r + \sum_{i' \in W_j} \mu_{i'} - r\right)(1 + \mu_i)/2 + r/2 - 0.5\sum_{i' \in W_j} \mu_{i'}.$$

The above expression can be simplified to get

$$\mathbb{E}[1/s_{ij} \,|\, y_j = 1] = r/2 + (1 - \mu_i^2)/2 + 0.5r\mu_i \frac{\sum_{i' \in W_j} \mu_{i'}}{|W_j|}.$$

We now make a few approximations based on the concentration of the sum of independent random variables around its mean. In particular, since $|W_j| = r \to \infty$, it follows (from say Chernoff bound) that

$$\frac{\sum_{i' \in W_j} \mu_{i'}}{|W_j|} \approx \mathbb{E}[\mu_i] = \mu,$$

where we use the notation $X \approx y$ to denote that the random variable $X$ concentrates around quantity $y$. We thus have that

$$\mathbb{E}[1/s_{ij} \,|\, y_j = 1] \approx r/2 + (1 - \mu_i^2)/2 + 0.5r\mu_i\mu.$$

In a similar fashion, in the case when $y_j = -1$, we can write (using lemmas 1 and 2)

$$\mathbb{E}[1/s_{ij} \,|\, y_j = -1]$$
$$= \left(2\mathbb{E}[d_j^+ \,|\, y_j = -1, w_i(t_j) = 1] - r\right)\Pr(w_i(t_j) = 1 \,|\, y_j = -1)$$
$$+ r - \mathbb{E}[d_j^+ \,|\, y_j = -1]$$
$$= \left(1 + \mu_i + r - \sum_{i' \in W_j} \mu_{i'} - r\right)(1 - \mu_i)/2 + r/2 + 0.5\sum_{i' \in W_j} \mu_{i'}$$
$$= r/2 + (1 - \mu_i^2)/2 + 0.5r\mu_i \frac{\sum_{i' \in W_j} \mu_{i'}}{|W_j|}.$$

As a result, given some assignment of true labels for the tasks and using the concentration approximation (as $r \to \infty$) $s_{ij} \approx 1/\mathbb{E}[1/s_{ij} \,|\, y_j = 1]$ when $y_j = 1$ and $s_{ij} \approx 1/\mathbb{E}[1/s_{ij} \,|\, y_j = -1]$ when $y_j = -1$, we have

$$\frac{1}{|\mathcal{T}_i|}\sum_{j \in \mathcal{T}_i} s_{ij} \approx \frac{1}{|\mathcal{T}_i|}\sum_{j \in \mathcal{T}_i} \frac{1}{r/2 + (1 - \mu_i^2)/2 + 0.5r\mu_i\mu} = \frac{1}{r/2 + (1 - \mu_i^2)/2 + 0.5r\mu_i\mu},$$

where $\mathcal{T}_i$ denotes the set of tasks that worker $w_i$ was assigned. Thus the penalty received by worker $w_i$ concentrates around $1/g(\mu_i)$, where the function $g(\cdot)$ is defined as $g(x) = r/2 + (1 - x^2)/2 + 0.5rx\mu$. Note that $g'(x) = -x + 0.5r\mu$. Thus, for $r$ large enough and $\mu > 0$, we must have that $g'(x) > 0$ for any $x \in [-1, 1]$. Thus, the function $g(\cdot)$ is increasing on the domain $[-1, 1]$, from which we can conclude that the penalty decreases with the increase in the reliability of the worker. The result of the theorem now follows.

## A.2 Proof of Theorem 2

The proof is similar to that of theorem 1 above. First, we need to compute $\mathbb{E}[d_j^+ \mid y_j = 1]$ for any task $t_j$. In particular, we can write

$$d_j^+ = \sum_{a \in A_j} \mathbf{1}[a(t_j) = 1] + \sum_{i \in H_j} \mathbf{1}[w_i(t_j) = 1]$$

$$= |A_j| + \sum_{i \in H_j} \mathbf{1}[w_i(t_j) = 1].$$

where $H_j$ and $A_j$ denotes the set of honest workers and adversaries who label task $t_j$. We now note that $\mathbb{E}[|A_j|] = (1 - q)r$ and $\mathbb{E}[|H_j|] = qr$, and following the sequence of arguments in lemmas 1 and 2 we can write

$$\mathbb{E}[d_j^+ \mid y_j = 1] \approx (1 - q)r + \frac{rq}{2}(1 + \mu).$$

In a similar fashion, we can show that

$$\mathbb{E}[d_j^+ \mid y_j = -1] \approx (1 - q)r + \frac{rq}{2}(1 - \mu).$$

Similarly, for any honest worker $h_i$ we can write

$$\mathbb{E}[d_j^+ \mid y_j = 1, w_i(t_j) = 1] \approx (1 - q)r + \frac{1 - \mu_i}{2} + \frac{rq}{2}(1 + \mu)$$

and

$$\mathbb{E}[d_j^+ \mid y_j = -1, w_i(t_j) = 1] \approx (1 - q)r + \frac{1 + \mu_i}{2} + \frac{rq}{2}(1 - \mu).$$

Note that $\mathbb{E}[d_j^+ \mid y_j = 1]$ and $\mathbb{E}[d_j^+ \mid y_j = -1]$ are respectively $\mathbb{E}[1/s_{aj} \mid y_j = 1]$ and $\mathbb{E}[1/s_{aj} \mid y_j = -1]$ for any adversary $a$ labeling on task $t_j$. We can now compute the penalty assigned by task $t_j$ to honest worker $h_i$. Following the sequence of steps above, we can write

$$\mathbb{E}[1/s_{ij} \mid y_j = 1] \approx \left( 2(1 - q)r + 1 - \mu_i + rq + rq\mu - r \right)(1 + \mu_i)/2 + r - (1 - q)r - rq/2 - rq\mu/2$$

$$= (1 - q)r(1 + \mu_i)/2 + (1 - \mu_i^2)/2 + rq/2 + rq\mu\mu_i/2$$

$$= (1 - \mu_i^2)/2 + r/2 + r\mu_i(q\mu + 1 - q)/2.$$

In a similar fashion, we can show that

$$\mathbb{E}[1/s_{ij} \mid y_j = -1] \approx \left( 2(1 - q)r + 1 + \mu_i + rq - rq\mu - r \right)(1 - \mu_i)/2 + r - (1 - q)r - rq/2 + rq\mu/2$$

$$= (1 - q)r(1 - \mu_i)/2 + (1 - \mu_i^2)/2 + rq/2 + rq\mu\mu_i/2$$

$$= (1 - \mu_i^2)/2 + r/2 + r\mu_i(q\mu - 1 + q)/2.$$

We thus have that the total penalty received by an honest worker is given by

$$\frac{1}{|\mathcal{T}_i|} \sum_{t_j \in \mathcal{T}_i} s_{ij} \approx \frac{\gamma}{(1 - \mu_i^2)/2 + r/2 + r\mu_i(q\mu + 1 - q)/2} + \frac{(1 - \gamma)}{(1 - \mu_i^2)/2 + r/2 + r\mu_i(q\mu - 1 + q)/2}$$

Similarly, the total penalty received by an adversary is given by

$$\frac{1}{|\mathcal{T}_a|} \sum_{t_j \in \mathcal{T}_a} s_{aj} \approx \frac{\gamma}{(1 - q)r + rq(1 + \mu)/2} + \frac{(1 - \gamma)}{(1 - q)r + rq(1 - \mu)/2}.$$

Now consider the special case in which $\gamma = 1/2$ and $\mu_i = 1$, where we get

$$\frac{1}{|\mathcal{T}_i|} \sum_{t_j \in \mathcal{T}_i} s_{ij} \approx \frac{1/(2r)}{1 - q/2 + q\mu/2} + \frac{1/(2r)}{q/2 + q\mu/2}$$

and

$$\frac{1}{|\mathcal{T}_a|}\sum_{t_j \in \mathcal{T}_a} s_{aj} \approx \frac{1/(2r)}{1 - q/2 + q\mu/2} + \frac{1/(2r)}{1 - q/2 - q\mu/2}.$$

In this special case it is easy to see that the penalty assigned to the adversaries is higher than the penalty assigned to the honest workers if and only if

$$q/2 + q\mu/2 > 1 - q/2 - q\mu/2 \iff q + q\mu > 1 \iff q > 1/(1+\mu).$$

Now consider the more general setting. We write

$$\frac{1}{|\mathcal{T}_i|}\sum_{t_j \in \mathcal{T}_i} s_{ij} \approx \frac{2\gamma/r}{(1 - \mu_i^2)/r + 1 + \mu_i(q\mu + 1 - q)} + \frac{2(1-\gamma)/r}{(1 - \mu_i^2)/r + 1 + \mu_i(q\mu - 1 + q)}$$

$$\leq \frac{2\gamma/r}{1 + \mu_i(q\mu + 1 - q)} + \frac{2(1-\gamma)/r}{1 + \mu_i(q\mu - 1 + q)}.$$

Similarly,

$$\frac{1}{|\mathcal{T}_a|}\sum_{t_j \in \mathcal{T}_a} s_{aj} = \frac{2\gamma/r}{2 - q + q\mu} + \frac{2(1-\gamma)/r}{2 - q - q\mu}.$$

More generally, the penalty of the honest worker is less than that of the adversary only if the reliability of the honest worker is "high enough". To simplify the expressions, we assume that $\gamma = \frac{1}{2}$:

$$\frac{1/2}{1 + \mu_i(q\mu + 1 - q)} + \frac{1/2}{1 + \mu_i(q\mu - 1 + q)} \leq \frac{1/2}{2 - q + q\mu} + \frac{1/2}{2 - q - q\mu}$$

$$\iff \frac{1 + \mu_i q\mu}{(1 + \mu_i q\mu)^2 - (1-q)^2\mu_i^2} \leq \frac{2 - q}{(2-q)^2 - q^2\mu^2}$$

$$\iff [(2-q)^2 - q^2\mu^2](1 + \mu_i q\mu) \leq (2-q)[(1 + \mu_i q\mu)^2 - (1-q)^2\mu_i^2] \leq (2-q)(1 + q^2\mu^2 + 2q\mu\mu_i)$$

$$\iff (2-q)^2 - q^2\mu^2 - (2-q)(1 + q^2\mu^2) \leq \mu_i(2q\mu(2-q) - qw[(2-q)^2 - q^2\mu^2])$$

$$\iff \mu_i \geq \frac{(2-q)(1 - q - q^2\mu^2) - q^2\mu^2}{(2-q)q + q^2\mu^2}.$$

It is easy to see that the RHS expression is always less than or equal to 1. Thus, depending on the choice of the parameters, some low reliable honest workers will receive higher penalty than the adversaries.

### A.3 Proof of Theorem 3

We prove the result for the the case when there exists at least one subset $\mathcal{T}' \subseteq \mathcal{T}$ such that $\mathrm{PreIm}(\mathcal{T}') \leq k$. Otherwise, the lower bound $L = 0$ by definition and the result of the theorem is trivially true.

Let $H^*$ denote the set $\mathrm{PreIm}(\mathcal{T}^*)$ where

$$\mathcal{T}^* \overset{\text{def}}{=} \underset{\mathcal{T}' \subseteq \mathcal{T}\,:\,|\mathrm{PreIm}(\mathcal{T}')| \leq k}{\arg\max} |\mathcal{T}'|.$$

For a given decision rule $R \in \mathcal{C}$, we now construct an adversary strategy $\sigma^*$ under which atleast $L$ tasks are affected for some ground-truth labeling. Specifically for a fixed honest worker labeling pattern and ground-truth labeling $\mathbf{y}$ of the tasks, consider the following adversary strategy: letting $H^* = \{h_1, h_2, \ldots, h_{|H^*|}\}$ and the set of adversaries $A = \{a_1, a_2, \ldots, a_k\}$, we have (recall the notation in Section 2)

$$a_i(t) = \begin{cases} -h_i(t), & \text{if } t \in \mathcal{T}^* \\ h_i(t), & \text{otherwise} \end{cases} \forall i = 1, 2, \ldots, |H^*|,$$

In other words, the adversaries label *opposite* to the honest workers in $H^*$ for tasks in $\mathcal{T}^*$ and agree with them for all other tasks. Note that since $|H^*| \leq k$ by construction, the above strategy is

feasible. In addition, if $|H^*| < k$, then we only use $|H^*|$ of the $k$ adversary identities and not use the remaining.

Now consider the scenario in which the true labels of all tasks in $\mathcal{T}^*$ are reversed, let this ground-truth be denoted by $\hat{\mathbf{y}}$. This would alter the labels of the honest workers (since we assumed they always label correctly). In particular, let $\tilde{h}(t)$ denote the label of an honest worker in this scenario. Then, we have that for any honest worker

$\tilde{h}(t) = h(t), \ \forall \ t \notin \mathcal{T}^*$ and

$$\tilde{h}(t) = -h(t) \ \forall \ t \in \mathcal{T}^*, \ \forall \ h \in H. \quad (1)$$

Correspondingly, according to the adversary labeling strategy $\sigma^*$ above, the adversary labels would also change. In particular, using $\tilde{a}(t)$ to denote the adversary label in this scenario, we have

$\tilde{a}(t) = a(t), \ \forall \ t \notin \mathcal{T}^*$ and

$$\tilde{a}(t) = -a(t) \ \forall \ t \in \mathcal{T}^*, \ \forall \ a \in A. \quad (2)$$

Finally, let $\tilde{\mathcal{L}}$ denote the labeling matrix corresponding to this new scenario. We now argue that $\tilde{\mathcal{L}} = P\mathcal{L}$ for some $n \times n$ permutation matrix $P$. In order to see this consider, for any worker $w$ (honest or adversary), let $r(w)$ and $\tilde{r}(w)$ respectively denote the row vectors in matrices $\mathcal{L}$ and $\tilde{\mathcal{L}}$. We show that $\tilde{\mathcal{L}}$ can be obtained from $\mathcal{L}$ through a permutation of the rows. For that, first note that for any honest worker $h \notin H^*$, we must have by definition of $\mathrm{PreIm}$ that $h(t) = 0$ for any $t \in \mathcal{T}^*$. Thus, it follows from (1) that $\tilde{h}(t) = -h(t) = 0 = h(t)$ for any $t \in \mathcal{T}^*$. Furthermore, $\tilde{h}(t) = h(t)$ for any $t \notin \mathcal{T}^*$ by (1). Therefore, we have that $r(h) = \tilde{r}(h)$ for any $h \notin H^*$. Now consider an honest worker $h_i \in H^*$ for some $i$. We now argue that $r(h_i) = \tilde{r}(a_i)$. To see this, for any task $t \notin \mathcal{T}^*$, we have by (2) that $\tilde{a}_i(t) = a_i(t) = h_i(t)$, where the second equality follows from our definition of the adversary strategy. Similarly, for any $t \in \mathcal{T}^*$, we have $\tilde{a}_i(t) = -a_i(t)$ by (2) and $a_i(t) = -h_i(t)$ (by the adversary strategy). Hence, we must have $\tilde{a}_i(t) = h_i(t)$ for any $t \in \mathcal{T}^*$. Thus, we have shown that the rows $\tilde{r}(a_i) = r(h_i)$ for any $i$. Thus, $\tilde{\mathcal{L}}$ is obtained from $\mathcal{L}$ by swapping rows corresponding to $h_i$ with $a_i$ for all $i$.

Now that we have shown that $\tilde{\mathcal{L}} = P\mathcal{L}$ for some permutation matrix $P$, it follows from the fact that $R \in \mathcal{C}$ that $R(\tilde{\mathcal{L}}) = R(\mathcal{L})$. Thus, the label assigned by $R$ to all tasks in $\mathcal{T}^*$ is the same under both scenarios. As a result, it follows that $\mathrm{Aff}(R, \sigma^*, \mathbf{1}) + \mathrm{Aff}(R, \sigma^*, \hat{\mathbf{y}}) = |\mathcal{T}^*| = 2 * L$ and therefore, either $\mathrm{Aff}(R, \sigma^*, \mathbf{y}) \geq L$ or $\mathrm{Aff}(R, \sigma^*, \hat{\mathbf{y}}) \geq L$. The result of the theorem now follows.

Also, note that the simple majority decision rule, with labels chosen randomly in case of a tie, achieves this lower bound.

### A.4 Proof of Theorem 4

Before we can prove the theorem, we need the following definitions and lemmas.

**Definition 2.** A bipartite graph $B = (U, V, E)$ is termed *degenerate* if the following condition is satisfied:
$$|U| > |V|$$

**Definition 3.** A bipartite graph $B = (U, V, E)$ is termed *growth* if the following condition is satisfied:
$$\forall \ U' \subseteq U \ \ |U'| \leq |\mathrm{Img}(U')|$$

**Lemma 3.** Any bipartite graph can be decomposed into degenerate and growth sub-graphs where there are cross-edges only from the growth component to the degenerate component.

*Proof.* Let $B = (U, V, E)$ be a given bipartite graph. Define $U^*$ to be the largest subset of $U$ such that $|U^*| > |\mathrm{Img}_B(U^*)|$ where $\mathrm{Img}_B$ denotes the image in the graph $B$. If no such $U^*$ exists then the graph is already growth and we are done. Else, we claim that the sub-graph (say $C$) of $B$ restricted to $U \setminus U^*$ on the left and $V \setminus \mathrm{Img}_B(U^*)$ on the right is growth. Otherwise,

there exists a subset $U'$ such that $|U'| > |\text{Img}_C(U')|$ where $\text{Img}_C(U') \subseteq V \setminus \text{Img}_B(U^*)$ denotes the image of $U'$ in the sub-graph $C$. But then, we can add $U'$ to $U^*$ to get a larger degenerate sub-graph in $B$ which contradicts our choice of $U^*$. To see this, consider the set $U^* \cup U'$ on the left and $\text{Img}_B(U^*) \cup \text{Img}_C(U')$ on the right. We have $|U^* \cup U'| = |U^*| + |U'| > |\text{Img}_B(U^*)| + |\text{Img}_C(U')| = |\text{Img}_B(U^* \cup U')|$. Also, note that the only cross-edges are from $U \setminus U^*$ to $\text{Img}_B(U^*)$. This shows that any bipartite graph can be decomposed into degenerate and growth sub-graphs with cross-edges only from the growth to the degenerate sub-graph. $\qquad\square$

**Lemma 4.** Let $B = (U, V, E)$ be any bipartite graph and suppose that $M$ is *any* semi-matching on $B$. Further, let $C = (U, V_1 \subseteq V, E_1)$ be any subgraph of $B$. Starting with $M_1 \subseteq M$, we can use algorithm $\mathcal{A}_{SM2}$ in [OPT] to obtain an optimal semi-matching $N$ on $C$. Let the nodes in $U$ be indexed such that $deg_M(1) \geq deg_M(2) \geq \ldots deg_M(|U|)$ and indexed again such that $deg_N(1) \geq deg_N(2) \geq \ldots deg_N(|U|)$. Then for any $1 \leq l \leq |U|$, we have $\sum_{i=1}^{l} deg_N(i) \leq \sum_{i=1}^{l} deg_M(i)$, i.e. the sum of the degrees of the top $l$ nodes in $U$ only decreases as we go from $M$ to $N$.

*Proof.* Note that if we restrict $M$ to just the nodes $V_1$, we get a feasible semi-matching $M_1$ on $C$. Algorithm $\mathcal{A}_{SM2}$ proceeds by the iterated removal of cost-reducing paths. Note that when a cost-reducing path is removed, load is transferred from a worker with larger degree to a worker with strictly smaller degree. To see this, let $P = (u_1, v_1, u_2, \ldots u_k)$ be a cost-reducing path. This means that $deg(u_1) > deg(u_k) + 1$. When we eliminate the cost-reducing path $P$, the degree of $u_1$ decreases by 1 and that of $u_k$ increases by 1, but still the new degree of $u_k$ is strictly lower than the old degree of $u_1$. In other words, if $d_1^B \geq d_2^B \geq \ldots d_{|U|}^B$ and $d_1^A \geq d_2^A \geq \ldots d_{|U|}^A$ be the degrees of the nodes in $U$ before and after the removal of a cost-reducing path, then $\sum_{i=1}^{l} d_i^A \leq \sum_{i=1}^{l} d_i^B$ for any $1 \leq l \leq |U|$. Since this invariant is satisfied after every iteration of algorithm $\mathcal{A}_{SM2}$, it holds at the beginning and the end and we have

$$\sum_{i=1}^{l} deg_N(i) \leq \sum_{i=1}^{l} deg_{M_1}(i) \tag{3}$$

However, note that when we restrict $M$ to only the set $V_1$, the sum of the degrees of the top $l$ nodes in $U$ can only *decrease*, i.e.

$$\sum_{i=1}^{l} deg_{M_1}(i) \leq \sum_{i=1}^{l} deg_M(i) \tag{4}$$

Combining equations 3 and 4, the result follows. $\qquad\square$

**Notation.** First, let $\mathcal{T}^+$ denote the set $\{t^+ : t \in \mathcal{T}\}$, and similarly $\mathcal{T}^-$ denote the set $\{t^- : t \in \mathcal{T}\}$. Now partition the set of task copies $\mathcal{T}^+ \cup \mathcal{T}^-$ into $E \cup F$ such that for any task $t$, if the **true** label is 1, we put $t^+$ in $E$ and $t^-$ in $F$, otherwise, we put $t^-$ in $E$ and $t^+$ in $F$. Thus, $E$ contains task copies with *true* labels while $F$ contains task copies with *incorrect* labels. Since honest workers always provide the true label, all honest workers have edges *only* to the set of task copies in $E$. However, adversaries can have edges to task copies in $E$ and $F$. In addition, it is easy to see that the bipartite graph on the honest workers $H$ and task copies in $E$ is the same as $\mathcal{B}_H$. As a result, the optimal semi-matching $M_E$ over the sub-graph from $H$ to $E$ is the same as the optimal semi-matching over the bipartite graph $\mathcal{B}_H$, which we denote by $M_H$. Thus, the degrees of the honest workers in $M_E$ are by hypothesis of the theorem $d_1, d_2, \ldots, d_{|H|}$. Without loss of generality, suppose that honest workers are indexed such that $d_1 \geq d_2 \geq \cdots \geq d_{|H|}$ and $d_h$ denote the degree of honest worker $h$.

## Adversary strategy that affects atleast $\sum_{i=1}^{k-1} d_i$ tasks

We now exhibit an adversary strategy that results in incorrect labels for at least $\sum_{i=1}^{k-1} d_i$ tasks. For a given honest worker labeling pattern, the adversaries target workers $\{1, 2, \ldots, k-1\}$ in $H$: for each honest worker $h$, the adversary labels opposite to $h$ on every task that $h$ is mapped to in the semi-matching $M_E$. Furthermore, the adversary uses the last identity $k$ to label opposite the true label for every task $t \in \mathcal{T}$ for which one of the first $k-1$ adversaries have not already labeled on. We now argue that under the penalty-based aggregation algorithm, this adversary strategy results in incorrect labels for at least $\sum_{i=1}^{k-1} d_i$ tasks. To see this, first note that the conflict set $\mathcal{T}_{cs}$ is the entire set of

tasks $\mathcal{T}$. The bipartite graph $\mathcal{B}^{cs}$ decomposes into two disjoint bipartite graphs: bipartite graph $\mathcal{B}_E$ from $H$ to $E$ and semi-matching $M_F$ from $A$ to $F$ that represents the adversary labeling pattern (it is a semi-matching because there is exactly one adversary labeling on each task). Since the bipartite graph $\mathcal{B}^{cs}$ decomposes into two disjoint bipartite graphs, computing the optimal semi-matching on $\mathcal{B}^{cs}$ is equivalent to separately computing optimal semi-matchings on $\mathcal{B}_E$ and $M_F$. Since $M_E$ is the optimal semi-matching on $\mathcal{B}_E$ and $M_F$ is already a semi-matching by construction, the optimal semi-matching of $\mathcal{B}^{cs}$ is the disjoint union of $M_E$ and $M_F$. It is easy to see that in the resultant semi-matching honest worker $i$ and adversary $i$ have the same degrees for $i = 1, 2, \ldots, k-1$. Hence, for every task that is labeled by honest worker $i$ for $i = 1, 2, \ldots, k-1$, the algorithm outputs label 0. Thus, this adversary strategy results in incorrect labels for at least $\sum_{i=1}^{k-1} d_i$ tasks.

**Upper bound U**

We assume in the arguments below that the optimal semi-matching in HARD PENALTY Algorithm is computed on the entire task set and not just the conflict set $\mathcal{T}_{cs}$ (We drop the $cs$ superscript and refer to this graph as just $\mathcal{B}$ below, note that $\mathcal{B}$ contains both real and fake copies of the tasks). However, the bounds provided still hold as a result of lemma 4 above. Also, we assume that the adversary labeling strategy is always a semi-matching, i.e. there is atmost one adversary label for any task. If the adversary labeling strategy is not a semi-matching, they can replace it with an alternate strategy where they only label for tasks to which they will be mapped in the optimal semi-matching (the adversaries can compute this since they have knowledge of the honest workers' labels). The optimal semi-matching doesn't change (otherwise it contradicts the optimality of the original semi-matching) and hence neither does the number of affected tasks.

We first state the following important lemma:

**Lemma 5.** For any adversary labeling strategy, let $\mathcal{B}(E)$ denote the bipartite graph $\mathcal{B}$ restricted to all the workers $W$ on the left and "real tasks" $E$ on the right, and $M$ be the optimal semi-matching on the bipartite graph $\mathcal{B}$. Further, let $M(E) \subset M$ be the optimal semi-matching restricted just to the task set $E$. Then, $M(E)$ is an optimal semi-matching for the sub-graph $\mathcal{B}(E)$.

*Proof.* Suppose the statement is not true and let $N(E)$ denote the optimal semi-matching on $\mathcal{B}(E)$. We use $d_w(K)$ to denote the degree of worker $w$ in a semi-matching $K$. Note that, $d_a(N(E)) \leq d_a(M(E)) \leq d_a(M)$ for all adversaries $a \in A$. For the adversaries who did not agree with any honest worker, they will have degrees 0 in the semi-matchings $N(E)$ and $M(E)$ but the inequality is still satisfied. Now, since $N(E)$ is an optimal semi-matching and $M(E)$ is not, we have that

$$cost(N(E)) < cost(M(E)) \Rightarrow$$

$$\sum_{h \in H} d_h(N(E))^2 + \sum_{a \in A} d_a(N(E))^2 < \sum_{h \in H} d_h(M(E))^2 + \sum_{a \in A} d_a(M(E))^2 \quad (5)$$

Now, consider the semi-matching $N$ on $\mathcal{B}$ where we start with the semi-matching $N(E)$ and then map the adversaries $A$ to the tasks in $F$ to which they were assigned in the original optimal semi-matching $M$. Now, we claim that $cost(N) < cost(M)$ which will be a contradiction since $M$ was assumed to be an optimal semi-matching on $\mathcal{B}$.

$$cost(M) - cost(N)$$
$$= \sum_{h \in H} d_h(M)^2 + \sum_{a \in A} d_a(M)^2 - \left( \sum_{h \in H} d_h(N)^2 + \sum_{a \in A} d_a(N)^2 \right)$$
$$= \sum_{h \in H} d_h(M(E))^2 + \sum_{a \in A} (d_a(M(E)) + \Delta_a)^2$$
$$- \left( \sum_{h \in H} d_h(N(E))^2 + \sum_{a \in A} (d_a(N(E)) + \Delta_a)^2 \right)$$

$$\text{where } \Delta_a \stackrel{\text{def}}{=} d_a(M) - d_a(M(E)) \geq 0$$

$$= \left( \sum_{h \in H} d_h(M(E))^2 + \sum_{a \in A} d_a(M(E))^2 - \sum_{h \in H} d_h(N(E))^2 - \sum_{a \in A} d_a(N(E))^2 \right)$$
$$+ 2 \sum_{a \in A} (d_a(M(E)) - d_a(N(E))) * \Delta_a > 0$$

$$\text{since } d_a(M(E)) \geq d_a(N(E)) \text{ as stated above} \quad (6)$$

Therefore, $M(E)$ is an optimal semi-matching for the sub-graph $\mathcal{B}(E)$. $\qquad \square$

**Part 1. Adversaries only provide incorrect labels or atmost 1 adversary provides correct labels**

First, we provide an upper bound on the number of affected tasks when the adversaries only *disagree* with the honest workers.

**Lemma 6.** Suppose that the adversaries never agree with the honest workers. Let $M$ be an arbitrary semi-matching on the bipartite graph $\mathcal{B}$ and suppose that this semi-matching is used in the PENALTY-BASED AGGREGATION Algorithm to compute the true labels of the tasks. Further, let $b_1 \geq b_2 \geq \cdots \geq b_{|H|}$ denote the degrees of the honest workers in this semi-matching where $b_i$ is the degree of honest worker $h_i$. Then, the number of affected tasks is atmost $\sum_{i=1}^{k} b_i$.

*Proof.* It follows from the assumption that adversaries never agree with the honest workers that there are no cross-edges between $A$ and $E$ in the bipartite graph $\mathcal{B}$. Thus, for any adversary labeling pattern, we can decompose the bipartite graph $\mathcal{B}$ into $\mathcal{B}_E$ and $\mathcal{B}_F$, where $\mathcal{B}_F$ is the bipartite graph from the adversaries $A$ to the task copies in $F$. This further means that the semi-matching $M$ is a disjoint union of semi-matchings on $\mathcal{B}_E$ and $\mathcal{B}_F$. Let the semi-matchings on the sub-graphs be termed as $M(E)$ and $M(F)$ respectively. Further, let $T \subseteq \mathcal{T}$ denote the set of tasks that are affected (receive incorrect labels) under this strategy of the adversaries and when the semi-matching $M$ is used to compute the reputations of the workers. We claim that $|T| \leq \sum_{i=1}^{k} b_i$. To see this, for each adversary $a \in A$, let $H(a) \subset H$ denote the set of honest workers who have "lost" to $a$ i.e., for each worker $h \in H(a)$ there exists a task $t \in \mathcal{T}$ such that $h$ is mapped to $t$ in $M(E)$, $a$ is mapped to $t$ in $M(F)$, and the degree of $h$ in $M(E)$ is greater than or equal to the degree of $a$ in $M(F)$. Of course, $H(a)$ may be empty. Let $\bar{A}$ denote the set of adversaries $\{a \in A : H(a) \neq \emptyset\}$ and let $\bar{H}$ denote the set of honest workers $\bigcup_{a \in \bar{A}} H(a)$. Now define a bipartite matching between $\bar{A}$ and $\bar{H}$ with an edge between $a \in \bar{A}$ and $h \in \bar{H}$ if and only if $h \in H(a)$. This bipartite graph can be decomposed into degenerate and growth sub-graphs by lemma 3. In the growth sub-graph, by Hall's condition, we can find a perfect matching from adversaries to honest workers. Let $(S_1, H_1)$ with $S_1 \subseteq \bar{A}$ and $H_1 = \text{Img}(S_1)$ be the degenerate component. The number of tasks that adversaries in $S_1$ affect is bounded above by $\sum_{h \in \text{Img}(S_1)} b_h$. Similarly, for $S_2 = \bar{A} \setminus S_1$, we can match each adversary to a *distinct* honest worker whose degree is greater than or equal to the degree of the adversary. We can now bound the number of affected tasks by the adversaries in $S_2$ by the sum of their degrees, which in turn is bounded above by the sum of the degrees of honest workers that the adversaries are matched to. Let $H_2$ denote the set of honest workers matched to adversaries in the perfect matching. Thus, we have bounded the number of affected tasks above by $\sum_{h \in H_1 \cup H_2} b_h$. It is easy to see that $|H_1 \cup H_2| \leq k$. Therefore, $\sum_{h \in H_1 \cup H_2} b_h \leq \sum_{i=1}^{k} b_i$. Therefore, the number of affected tasks $|T|$ is atmost $\sum_{i=1}^{k} b_i$ if the adversaries only disagree with the honest workers. $\qquad\square$

Note that since the above lemma is true for *any* choice of semi-matching $M$ it is true in particular for the optimal semi-matching on $\mathcal{B}$. Therefore, it gives us an upper bound on the number of affected tasks when the adversaries only disagree with the honest workers.

Now, consider the case when there is exactly 1 adversary that agrees with the honest workers and all other adversaries only disagree. Let $M$ be the optimal semi-matching on the bipartite graph $\mathcal{B}$ resulting from such an adversary strategy and let $a$ denote the adversary who agrees with the honest workers. Observe that we can apply the argument in lemma 6 above to get a bound on the number of affected tasks that the adversaries "win" against the honest workers. Let $T_1$ denote the set of these tasks. There are two possible scenarios: either we obtain a perfect matching between the $k$ adversaries and some $k$ honest workers (refer to the proof above). In this scenario, we have accounted for all the affected tasks in the original semi-matching $M$. In the other scenario, when the degenerate component is non-empty, we have a total of at most $k - 1$ honest workers on the right and we bound $T_1$ by the sum of the degrees of these honest workers. Note, however that we may be missing out on some of the affected tasks, namely those that the adversary $a$ "loses" against *other adversaries*. The tasks that we might be missing out on correspond exactly to the tasks in $E$ that the adversary $a$ is mapped to in the semi-matching $M$. Specifically, let $M(E)$ denote the semi-matching $M$ restricted to just the real tasks $E$. Then it follows that we can bound the number of affected tasks by $|T_1| + d_a(M(E))$ where $d_a(M(E))$ denotes the degree of $a$ in $M(E)$.

Next observe that in both cases, we have bounded the number of affected tasks by the sum of the degrees of some $k$ workers in the semi-matching $M$ restricted to workers $H \cup \{a\}$ on the left and tasks $E$ on the right, i.e. in the semi-matching $M(E)$. Lemma 5 tells us that $M(E)$ is in fact, the optimal semi-matching on the subgraph from workers $H \cup \{a\}$ to tasks $E$. Finally, lemma 4 implies that this sum is atmost $\sum_{i=1}^{k} d_i$ (by starting with $M_H$ as a feasible semi-matching) and the bound follows.

**Part 2. Adversaries can provide arbitrary labels**

Now, consider the general case when any number of adversaries can agree with the honest workers. We further make the assumption that $|H| \geq 2 * |A|$, otherwise the upper bound below becomes $\sum_{i=1}^{|H|} d_i$, which is the set of all tasks $\mathcal{T}$ and is a trivial upper bound.

First recall that lemma 6 was applicable to any semi-matching and in fact, we can use the same argument even when the adversaries agree with the honest workers. Formally, let the set of affected tasks $T$ for an arbitrary adversary labeling strategy, resulting in an optimal semi-matching $M$ on the bipartite graph $\mathcal{B}$ be such that $T = T_H \cup T_A$ where $T_H$ are the tasks that the adversaries "win" against the honest workers and $T_A$ are the tasks that are affected when 2 adversaries are compared against each other in the final step of the PENALTY-BASED AGGREGATION Algorithm. We can apply the same argument as in lemma 6 to bound $|T_H|$ by the sum of the degrees of the top $k$ honest workers in the optimal semi-matching $M$. Further, we can bound $T_A$ by the number of tasks in $E$ that were mapped to adversaries in the optimal semi-matching $M$. This, in turn is equal to the sum of the degrees of the adversaries in the semi-matching $M$ restricted to just the real tasks $E$. Let $A_H \subseteq A$ denote the set of adversaries that are mapped to tasks in $E$ in the optimal semi-matching $M$. Therefore, we can bound the number of affected tasks $|T|$ by the sum of the degrees of the top $j = k + |A_H|$ workers in the semi-matching $M$ restricted to just the real tasks $E$. Now, we claim that this is upper bounded by the sum of the degrees of the top $j$ honest workers in the optimal semi-matching $M_H$ on the original honest worker sub-graph $\mathcal{B}_H$. To see this, start with $M_H$ as a feasible semi-matching from workers $H \cup A_H$ to tasks $E$. Lemma 4 tells us that the sum of the degrees of the top $j$ workers in the optimal semi-matching is atmost the sum of the degrees of the top $j$ honest workers in $M_H$. Further, lemma 5 tells us that the optimal semi-matching on the sub-graph from workers $H \cup A_H$ to tasks $E$ is precisely the semi-matching $M$ restricted to the real tasks $E$. This shows that we can bound the number of affected tasks by $\sum_{i=1}^{j} d_i$. Finally, note that $|A_H| \leq k \Rightarrow j \leq 2k$ and hence, we can bound the number of affected tasks by $\sum_{i=1}^{2k} d_i$.

**Uniqueness of degree-sequence in optimal semi-matchings**

In the arguments above, we have implicitly assumed some sort of uniqueness for the optimal semi-matching on any bipartite graph. Clearly its possible to have multiple optimal semi-matchings for a given bipartite graph. However, we prove below that the degree sequence of the vertices is unique across all optimal semi-matchings and hence our bounds still hold without ambiguity.

**Lemma 7.** Let $M$ and $M'$ be two optimal semi-matchings on a bipartite graph $B = (U, V, E)$ with $|U| = n$ and let $d_1 \geq d_2 \cdots \geq d_n$ and $d_1' \geq d_2' \geq \cdots d_n'$ be the degree sequence for the $U$-vertices in $M$ and $M'$ respectively. Then, $d_i = d_i' \ \forall \ 1 \leq i \leq n$, or in other words, any two optimal semi-matchings have the same degree sequence.

*Proof.* Let $l$ be the smallest index such that $d_l \neq d_l'$, note that we must have $l < n$ since we have that $\sum_{j=1}^{n} d_j' = \sum_{j=1}^{n} d_j$. This means that we have $d_j = d_j' \ \forall j < l$. Without loss of generality, assume that $d_l' > d_l$. Now, $\exists q \in \mathbb{N}$ such that $d_l'^q > d_l^q + \sum_{j=l+1}^{n} d_j^q$ and since $d_j = d_j' \ \forall j < l$, we have that $\sum_{j=1}^{n} d_j'^q \geq \sum_{j=1}^{l} d_j'^q > \sum_{j=1}^{n} d_j^q$. But, this is a contradiction since an optimal semi-matching minimizes the $L^p$ norm of the degree-vector for any $p \geq 1$ (Section 3.4 in [OPT]). Hence, we have that $d_i = d_i' \ \forall i$. $\qquad\square$

**References**

[OPT] N. J. Harvey, R. E. Ladner, L. Lovász, and T. Tamir. Semi-matchings for bipartite graphs and load balancing. In *Algorithms and data structures*, pages 294-306. Springer, 2003.