[Reviews · NeurIPS 2014]

Submitted by Assigned_Reviewer_6

This work explores the problem of identifying and filtering out *adversarial* workers in crowdsourced classification problems. A worker is adversarial if his/her voting strategy doesn’t follow the standard random process: vote correctly with probability p and incorrectly with probability 1-p, where p is the worker’s reliability. To solve the problem of adversarial worker filtering, this paper proposes to attach reputation values to workers. The reputation value of a worker is calculated based on if this worker’s votes agree with other workers’ (in particular, this paper proposes to penalize disagreements instead of awarding agreements) The authors propose two versions of the penalty algorithms and prove their properties with different adversarial worker voting strategies. Experiments on both synthetic and real datasets are conducted.

I think this paper addresses a very interesting problem. Most of other papers on label aggregation assume workers’ votes are i.i.d. samples from some distribution. This paper considers different voting strategies and takes a step towards a more realistic setting. However, the theoretical analysis on hard penalty algorithms are not too satisfactory since 1) it’s hard to interpret the results and 2) the authors make a strong assumption in their analysis (i.e., honest workers are perfect). I also have some concerns about the experiment results. (Please see comments below.)

Some detailed comments:

- In the analysis of hard penalty assignment algorithm (and also the simulation), you assume the honest workers are perfect, i.e., give correct votes with probability 1. This seems to be a fairly strong assumption to me. Is this necessary in your analysis? Do you have intuitions on how to relax this assumption?

- The results of Theorem 3 and 4 are represented using the workers’ voting graph B_H. It would help the readers to get intuitions by giving examples to illustrate the bounds.

- In the experiments results on Table 2, the number of users being filtered out is different in each setting. I think you should show all the results (i.e., set the number of filtered users from 1 to 10) since you won’t know the optimal parameters when running your algorithm in real applications.

- In the experiments, your algorithm boosts the performance of ITER (by Karger et. al. 2011) the most. However, ITER assumes all workers complete the same number of tasks. If you want to apply ITER in datasets without this property, you should modify their algorithm by normalizing the messages at every step during the message passing. Otherwise the algorithm performance would just be dominated by the accuracies of workers who complete the most number of tasks. Without this modification, your algorithm is essentially just filtering out low-accuracy workers completing lots of tasks. It doesn’t provide evidences that your algorithm can filter out adversarial workers.

AFTER THE REBUTTAL:

Thanks for addressing my concern about the ITER implmentation. I have updated my scores. Please report the updated results if accepted.

Overall, I think this is a nice paper addressing a more reasonable setting. I am still a little bit bothered by the perfect-worker assumption, but the empirical evaluations mitigate this shortcoming. I would vote for accept.
Summary: This paper addresses a very interesting problem, but the contribution is limited because of the strong assumption. I also have some concerns about the experiments. Overall, I think this is a borderline paper which I don’t have preferences on either accepting or rejecting.

Submitted by Assigned_Reviewer_15

The paper presented an approach to filter-out different adversarial voters in crowd sourcing systems in order to increase the quality of aggregation voting. The paper presents two different algorithm, soft and hard penalty assignment in order to filter out adversarial voters. Theoretical results justifies the proposed algorithms and the experimental results sounds promising.

This is an interesting problem and it would be nice to have more details about the applications of this problem.

It would be nice to have a few sentences explaining optimal semi-matching algorithm as it is being used as one of the important components in hard penalty assignment algorithm.

Why the hard penalty assignment performs consistently better than soft penalty assignment in real data set comparing to synthetic data set? This consistency is not clear for me.
Summary: The problem is interesting and the paper has been written well. It would be nice to have more discussion about the applications where the problem fits.

Submitted by Assigned_Reviewer_41

This paper looks at the problem of learning the true labels of objects in the context of crowd-sourcing systems and considers broader classes of adversarial voting strategies (and very powerful adversaries) than had been considered before.
Here, the authors propose novel reputation-based algorithms based on user disagreements and the use of semi-matchings which identify adversarial users. For these algorithms, the authors were able to show that they improve vote aggregation quality for 3 widely used vote aggregation algorithms. In addition, the authors show that their definition of reputation is compatible with the notion or reliability of users and that the algorithm can detect adversarial users (even when ‘smart’ adversarial strategies are employed), under the assumption that non-adversarial users have high reliability.

Finally, the authors establish bounds on the minimum damage that may be caused by smart adversaries.

Overall, I found the paper to be both interesting and original in that it considers different classes of adversaries than I had seen previously. At the same time, the authors make a number of strong assumptions which may not be realistic, such as (a) the degree to which ‘honest’ users label objects correctly (see Assumption 1 in Section 3.2), (b) the knowledge of the complete honest user voting pattern by adversarial users and (c) the knowledge of the decision rule being employed by the adversaries. Given these, it is not clear if the results of Theorem 3 and 4 do really have much practical relevance.

The other point I was curious about was the way the authors motivate not giving *any* credit for agreement (Section 2), since they didn’t want to give any incentive for adversaries to agree with honest users. In a scenario where all votes by honest users are known, this may be sensible (adversaries can simply agree with a set of items whose label they don’t want to influence and which have a large number of honest votes already), but in practical scenarios where adversaries don’t know the voting pattern, it’s not at all clear that no credit should be given for agreement.

Detailed comments:
- It would be very interesting to see how the results in Section 4 depend on the threshold of 20% placed on the set of users that are being removed.
Summary: An original look at user-filtering in the context of crowd-sourcing. The authors show good practical results and have some interesting theory behind it, but require some strong assumptions for their theoretical results.
Author Feedback
Author rebuttal: We thank the reviewers for their thoughtful and insightful comments and we briefly address them below:

HARD PENALTY ALGORITHM
Two of the reviewers point out that the assumptions made for the analysis of the hard penalty algorithm may be restrictive. We address these concerns as follows:

1. Analysis under the standard assumptions (honest voters randomly make mistakes): The theoretical results (Theorems 1 & 2) proved for the soft penalty algorithm under the standard settings extend readily to the hard penalty algorithm: penalty assigned by the hard-penalty algorithm is inversely proportional to the reliability of the user, so that the algorithm filters out users in the reverse order of their reliabilities. Specifically, if we use a random semi-matching (constructed by mapping each object to a user voting for the object uniformly at random), then the penalty assigned by the hard-penalty algorithm to user i concentrates (as the number of users and objects goes to infinity) around D_i g(w_i), where D_i is the degree (# of objects voted on) and w_i is the reliability of user i, and g(.) is a decreasing function. The proof of this result follows almost immediately from the arguments of Theorem 1 and 2.

2. Theorem 3 does NOT require ANY assumptions on the honest voting pattern. The proof of the lower bound in Theorem 3 relies on the argument of “indistinguishability”, which readily extends to the case when honest voters make mistakes.

3. The proof of Theorem 4 requires the assumption that honest voters make no mistakes for analytical tractability. Without this, proving theoretical guarantees in crowd-sourced settings even without adversaries for general graph structures is notoriously hard (see [9]). Our work has an added complication due to the presence of adversaries. Empirically, our hard (and soft-) penalty algorithm improves existing algorithms (Majority, EM and ITER) even when honest voters commit mistakes (see Table 2). If required, we can present additional empirical results and extend our theory bounds for special graph structures.

4. Evidence for sophisticated adversaries: Our assumption of sophisticated adversaries in Theorems 3 and 4 is common in the security community. This assumption also holds in practice as demonstrated by several real world examples: [1] shows rating manipulation using fake accounts on Digg.com; [12, 14] show examples of malicious activity in crowd-sourced systems; Molavi et al. show rating manipulations on Yelp; Mukherjee et al. show fake review groups that collaboratively target products in Amazon.

THEOREMS 3 & 4 RELATIONSHIP AND INTERPRETABILITY:
We apologize for the lack of clarity in exposing the tight relationship between Theorem 3 and Theorem 4 bounds in the paper. It is easy to show that when the top k users have distinct semi-matching degrees, then the lower bound L in Theorem 3 satisfies L >= (d_1 + d_2 + ... + d_{k-1})/2. This establishes the near optimality of Theorem 4 (upto constant factor) even under worst-case adversary settings.
Also, the bounds themselves become immediately interpretable when, for instance, the honest-user voting graph is r-right regular and an (alpha, beta) vertex expander from the objects to the users (the number of neighbors of every set S of objects of size at most (alpha m) is at least (beta |S|): here L >= k/(2r) and L <= k/(2 beta) provided alpha > k/(m beta), where L is as defined in Theorem 3. If the expander is randomly constructed, then asymptotically, r and beta will be ``close'' to each other, so that our bounds become tight. In addition, for the upper bound in theorem 4, we have that d_1 + … + d_2k <= 2k d_1 <= 2k/beta. Thus, for the case of a regular expander, the upper and lower bounds are tight (up to constants).

EXPERIMENTAL RESULTS
1. One of the reviewers points out that the ITER algorithm may be improved by normalizing the messages by the degrees of the nodes. We implemented this version of the algorithm (which we call ITER+). ITER+ outperforms ITER, but the hard-penalty algorithm still improves the performance of ITER+ consistently (with ~8% increase for the task2 dataset). We will report these results as well in the final version of the paper.
2. Threshold dependence (removing 20% of users for simulations and removing upto 10 users for the real-world datasets): One of the challenges in our algorithms is deciding how many users to remove in the filtering stage, which clearly depends on the setting. The results qualitatively remain the same upto removing 25% of the users in the simulations. For the real datasets, our algorithm matches or improves the accuracy for most cases when we remove upto 10 users.

RELATIONSHIP BETWEEN SOFT AND HARD PENALTY ALGORITHMS
Soft (hard) penalty algorithm performs better when the adversaries tend to have lower (higher) degrees. Theoretically, we can show that (see point 1 in HARD PENALTY ALGORITHM and Algorithm 2.1) the soft penalty algorithm normalizes the penalty by the degree of the user whereas the hard penalty does not, which implies that soft penalty is less influenced by user degrees. Empirically, for the real-world datasets, the un-normalized soft-penalty algorithm had a performance similar to the hard-penalty algorithm; we did not report these results due to space constraints. For simulations, soft (hard) performs better when the low (high) degree nodes are adversaries. Note that in Table 1, MV(soft) and MV(hard) had the same performance for uniform-high strategy while ITER and ITER(hard) had the same performance for the malicious-high strategy. These insights will be added to the final version.

CREDIT FOR AGREEMENT(Rev 41)
The soft-penalty algorithm does credit agreements by assigning a penalty equal to 1/(number of users who agree with the user) for each object. Since hard and soft-penalty are closely related to each other (see above), the hard-penalty algorithm (with a random semi-matching) also implicitly credits agreements.